# Genotypic and Allelic Frequencies of GJB2 Variants and Features of Hearing Phenotypes in the Chinese Population of the Dongfeng-Tongji Cohort

**DOI:** 10.3390/genes14112007

**Published:** 2023-10-27

**Authors:** Lanlai Yuan, Xiaohui Wang, Xiaozhou Liu, Sen Chen, Weijia Kong, Meian He, Yu Sun

**Affiliations:** 1Department of Otorhinolaryngology, Union Hospital, Tongji Medical College, Huazhong University of Science and Technology, Wuhan 430022, China; d202181855@hust.edu.cn (L.Y.); d202281891@hust.edu.cn (X.W.); d202181808@hust.edu.cn (X.L.); senchen@hust.edu.cn (S.C.); entwjkong@hust.edu.cn (W.K.); 2Institute of Otorhinolaryngology, Union Hospital, Tongji Medical College, Huazhong University of Science and Technology, Wuhan 430022, China; 3Department of Occupational and Environmental Health, Ministry of Education and State Key Laboratory of Environmental Health (Incubating), School of Public Health, Tongji Medical College, Huazhong University of Science and Technology, Wuhan 430030, China

**Keywords:** hearing loss, *GJB2* variants, c.109G>A, c.235delC, audiogram shapes

## Abstract

Background: This study aimed to describe the distribution of the genotype and allele frequencies of *GJB2* variants in the Chinese population of the Dongfeng Tongji cohort and to analyze the features of the hearing phenotype. Methods: We used data from 9910 participants in the Dongfeng Tongji cohort in 2013 and selected nine *GJB2* variants. Pure tone audiometry was employed to measure hearing. Differences in genotype and allele frequencies were analyzed via chi-squared test or Fisher’s exact test. Results: Of the 9910 participants, 5742 had hearing loss. The genotype frequency of the *GJB2* variant c.109G>A was statistically significantly distributed between the normal and impaired hearing groups, but not for the variant c.235delC. A higher frequency of the c.109G>A homozygous genotype was found in the hearing loss group (0.5%) than in the normal hearing group (0.1%). Patients with c.109G>A and c.235delC homozygous mutations exhibited varying degrees of hearing loss, mainly presenting sloping and flat audiogram shapes. Conclusions: A significant difference was found in the genotype frequency of the GJB2 variant c.109G>A between the case and control groups, but not in that of the variant c.235delC. Different degrees of hearing loss and various audiogram shapes were observed in patients with c.109G>A and c.235delC homozygous mutations.

## 1. Introduction

Hearing loss is a common sensory disorder affecting more than 1.5 billion people worldwide, including 430 million with moderate or severe hearing loss [1,2]. Hereditary factors or predispositions account for approximately 50% of hearing loss [3,4]. Genetic hearing loss includes syndromic and non-syndromic types, with marked genetic heterogeneity [5,6]. Of the genetic cases, about 70% are non-syndromic deafness, of which around 80% are inherited as an autosomal recessive trait [4]. To date, 124 genes have been identified for non-syndromic hearing loss and 77 for autosomal recessive non-syndromic hearing loss [7]. The most common cause of non-syndromic hearing loss is mutations in the gap junction protein beta 2 (*GJB2*) gene (OMIM: 121011), which encodes the gap junction protein connexin 26 (Cx26) on chromosome 13q12.11 [8,9,10]. 

Since hearing loss related to *GJB2* was first described in 1997 [11,12], more than 300 pathogenic variants of *GJB2* have been reported [10,12]. In different regions of the world, the spectrums and frequencies of mutations in *GJB2* vary widely [12]. The mutant allele of c.35delG is prevalent in populations of Europe and the Middle East, c.235delC is common in East Asia, and p.W24X is mainly found in India [12]. In addition, c.167delT and p.R143W are found in Ashkenazim and Ghanaians, respectively [12]. The c.109G>A, a specific variant of *GJB2*, is also common among deaf patients in East Asia [12]. However, the classification of this missense variant has been controversial [13]. The c.109G>A variant was initially reported as a benign polymorphism because of its high prevalence among people with normal hearing [14]. Later, c.109G>A was identified as homozygous or in trans with known pathogenic *GJB2* mutations in affected individuals [13,15,16]. Currently, c.109G>A is considered a pathogenic mutation with incomplete penetrance [13], and the homozygous mutation is associated with various hearing phenotypes [10,17]. Similarly, the other *GJB2* variants show diverse phenotypes, ranging from mild to profound [8,18], which largely depend on the genotype [8,10,19]. Patients with biallelic “truncating” mutations that completely block protein expression generally have severe to profound hearing loss, whereas patients with at least one “non-truncating” mutant allele have a lower hearing threshold due to impaired but not inactivated protein function [8,10,19]. However, even among patients with the same genotype of the *GJB2* mutation, auditory phenotypes show great variability [8,10,20]. Therefore, further investigation of genotype–phenotype correlations is needed to guide clinical evaluation and genetic counseling of patients with hearing loss associated with the *GJB2* mutations. 

Given the dominant role of *GJB2* mutations in hearing loss and their genetic heterogeneity and phenotypic diversity, it is important to analyze the frequency distribution and genotype–phenotype correlation of *GJB2* mutations in hearing loss patients for genetic diagnosis and counseling. This study aimed to: (1) analyze the frequencies of genotypes and alleles of *GJB2* mutations in the Chinese population from the Dongfeng Tongji (DFTJ) cohort and (2) investigate audiogram shapes in patients with hearing loss with regard to different genotypes of c.109G>A and c.235delC mutations in the *GJB2* gene.

## 2. Materials and Methods

### 2.1. Study Population 

The DFTJ cohort is a prospective study launched in September 2008 in Shiyan City, Hubei Province, China, based on retired employees of Dongfeng Motor Corporation [21]. The baseline survey was conducted between September 2008 and June 2010, with subsequent follow-ups every five years [21]. Information on demographic characteristics, lifestyle, health status, and medical history was collected by trained interviewers using a standardized questionnaire. In addition, participants performed physical examinations and provided samples of peripheral venous blood after an overnight fast. A total of 27,009 participants completed baseline questionnaires, physical examinations, and blood sample collections. Between April and October 2013, 25,978 participants completed the first follow-up. Pure tone audiometry (PTA) was first added in 2013, with 11,513 participants in the baseline population undergoing the test [22]. The approval of the DFTJ cohort protocol was obtained from the Medical Ethics Committee of the School of Public Health, Tongji Medical College, Huazhong University of Science and Technology, and Dongfeng General Hospital, Dongfeng Motor Corporation [21]. Written informed consent was obtained from each participant.

After the exclusion of 57 participants with invalid audiometric data and 1544 participants who lacked the genotype and failed quality control, this study included 9912 participants with complete data on nine *GJB2* mutations and audiometry. After those participants missing the age variable data (n = 2) were removed, 9910 participants were used for the final analysis.

### 2.2. Measurement of Hearing

In a special soundproof room at Dongfeng General Hospital, participants underwent pure tone tests administered by professional audiologists using a calibrated pure tone audiometer (Micro-DSP ZD21, Micro DSP Technology Co., Ltd., Chengdu, China) after an otological examination. Air conduction thresholds at 0.5, 1, 2, 4, and 8 kHz were recorded for each ear and coded as a maximum when the maximum was unresponsive [23]. Hearing loss was defined as an average threshold at 0.5, 1, 2, and 4 kHz (PTA_0.5–4 kHz_) greater than 25 decibels of hearing level (dB HL) in the better ear [24]. The severity of hearing loss was categorized as mild (25 dB HL < PTA_0.5–4 kHz_ ≤ 40 dB HL), moderate (40 dB HL < PTA_0.5–4 kHz_ ≤ 60 dB HL), and severe–profound (PTA_0.5–4 kHz_ > 60 dB HL). 

Based on Liu’s [20] classification criteria, the audiogram shapes of 5742 participants with hearing loss were classified as follows: sloping, where the difference between the mean thresholds at 4 kHz and 8 kHz and the mean thresholds at 0.5 kHz and 1 kHz was greater than 15 dB HL; flat, where the difference between the thresholds at 0.5 to 8 kHz was less than 15 dB HL; mid-frequency U-shaped, where the difference between the worst hearing thresholds at mid-frequency and those at lower and higher frequencies was greater than 15 dB HL; ascending, where the difference between the low-frequency thresholds and the high-frequency thresholds was greater than 15 dB HL; and residual, where only residual hearing was present at lower frequencies. Those not belonging to any of the above audiogram shapes were classified as “other”. Due to the relatively small sample size of patients with mid-frequency U-shaped (0.8%) and ascending (3.7%) hearing loss, the two types were combined into the “special” type. 

### 2.3. Genotyping

For the purpose of genetic analysis related to hearing loss, nine common GJB2 mutations in the Chinese population were selected. Nine *GJB2* variants or single nucleotide polymorphisms were selected as follows: c.11G>A (rs111033222), c.79G>A (rs2274084), c.*84T>C (rs3751385), c.109G>A (rs72474224), c.235delC (rs80338943), c.341A>G (rs2274083), c.368C>A (rs111033188), c.571T>C (rs397516878), and c.608T>C (rs76838169). These variants were genotyped in a genome-wide association study scan of the DFTJ cohort using Affymetrix Genome-Wide Human SNP Array 6.0 chips (Affymetrix China Inc., Shanghai, China)and Illumina Infinium Omni Zhong Hua-8 Chips (Illumina Trading Co., Ltd., Shanghai, China) [25]. In brief, these SNPs were successfully genotyped after quality control, excluding any SNPs with missing call rate > 5%, minor allele frequency < 1%, and *p* value of Hardy–Weinberg equilibrium < 10^−5^. There were 703,302 SNPs in the Illumina dataset and 549,196 in the Affymetrix dataset. Genotypes of markers in the Illumina and Affymetrix datasets were imputed separately using the 1000 Genomes Project ALL Phase 3 Integrated Release Version 5 haplotypes as a reference panel. Detailed descriptions of the genotyping and quality control processes are available elsewhere [26,27].

### 2.4. Statistical Analysis

The categorical variables were expressed as numbers with percentages, and the continuous variables as means with standard deviations. Comparisons of differences in genotype and allele frequencies between groups were conducted using the chi-squared test or Fisher’s exact test. The threshold for significance was set at a two-sided *p*-value of less than 0.05. All statistical analyses were carried out using R software (version 4.2.3, R Foundation for Statistical Computing, Vienna, Austria).

## 3. Results 

### 3.1. Basic Characteristics of the Study Population

Table 1 shows the characteristics of the total samples. A total of 9910 participants were included in the study, with a gender makeup of 52.4% (n = 5197) female and 47.6% (n = 4713) male. The mean age was 67.2 years. In total, 5742 participants had hearing loss, including 66.3% (n = 3806) mild, 27.5% (n = 1581) moderate, and 6.2% (n = 355) severe to profound. In addition, the mean ages of the controls and patients were 63.9 (7.0) and 69.7 (7.1) years, respectively.

### 3.2. Frequencies of the Genotypes and Alleles of the Mutations in the GJB2 Gene

Table 2 shows the genotype and allele frequencies of the nine mutations of the *GJB2* gene in the cohort and between the hearing-impaired and normal groups. Based on the variant annotation in the ClinVar database (https://www.ncbi.nlm.nih.gov/clinvar/, accessed on 13 May 2023), two variants are reported as pathogenic variants (c.109G>A and c.235delC); five are benign or likely benign variants (c.79G>A, c.*84T>C, c.341A>G, c.368C>A, and c.608T>C); one is uncertain (c.571T>C); and the last has a conflicting interpretation of pathogenicity (c.11G>A). We found a significant difference (*p*-value < 0.001) in the genotype frequency distribution of c.109G>A between the case and control groups (Table 2). Compared to the normal hearing group, the frequency of homozygous genotype was higher in the hearing loss group (0.1% in the normal hearing group and 0.5% in the hearing loss group). The homozygous mutation in c.109G>A affected a wide range of frequencies, and the detailed characteristics and pure tone thresholds of the 31 participants with homozygous genotypes in the cohort are shown in Table 3. The frequency distribution of p.V37 I alleles was nearly the same between the case and control groups (6.0% in controls and 6.3% in the hearing loss group), but not significantly different. Furthermore, the frequency of the c.109G>A homozygous genotype increased with hearing threshold in 5742 participants with hearing loss (0.4% mild, 0.7% moderate, and 0.8% severe–profound) (Figure 1), but no significant difference was observed between groups (*p* = 0.369). Likewise, no statistical difference was found for the frequency distribution of c.109G>A alleles across hearing loss severity groups (*p* = 0.274) (Figure 2).

However, regarding c.235delC, the most common mutation reported in the Chinese population with hearing loss, the frequency distributions of its genotype and allele were not found to differ significantly between controls and cases (Table 2). The carrier frequency of c.235delC was 1.1% in the general population controls (Table 2). Among the 5742 patients with hearing loss, the carrier frequency was 1.4%, and only two patients were identified with homozygous mutations (Table 2), with proportions of 0.1% and 0.3% in moderate and severe–profound hearing loss, respectively (Figure 1). Additionally, there was no significant difference in the allele frequency of c.235delC between the different hearing loss severity groups (*p* = 0.661) (Figure 2).

Moreover, among the five benign variants, we found statistically significant differences in the genotype and allele frequencies of c.*84T>C between case and control groups (Table 2). The allele frequency of c.*84T>C was comparable between the case and control groups (53.9% in the control group and 52.5% in the hearing loss group). A similar distribution was observed for the other four benign variants (c.79G>A, c.341A>G, c.368C>A, and c.608T>C) (Table 2). There were no statistical differences in genotype and allele frequencies for the remaining two variants (c.11G>A and c.571T>C) (Table 2).

### 3.3. Distribution of Audiogram Shapes for the Different Genotypes of rs72474224 (c.109G>A) and rs80338943 (c.235delC)

Figure 3a shows the distribution of audiogram shapes in the overall hearing loss group: 66.6% were sloping, 11.6% were flat, 4.5% were special, 6.7% were residual, and 10.6% were other. We analyzed the audiogram shape distributions of the two pathogenic variants (c.109G>A, and c.235delC) under different genotypes. As shown in Figure 3b, the c.109G>A and c.235delC variants presented various audiogram shapes, mainly showing sloping and flat types. The proportions of the sloping type in the c.109G>A heterozygous and homozygous genotypes were 70% and 82.1%, and the proportions of the flat type were 10% and 7.1%. For the c.235delC variant, the proportions of the sloping phenotype were 66% in the heterozygous genotypes and 100% in the homozygous genotypes; the flat phenotype was 14.4% in the heterozygous genotype.

## 4. Discussion 

In this study of 9910 participants from the DFTJ cohort, we described an overview of the genotype and allele frequency distributions of nine variants with the *GJB2* gene and analyzed the audiogram phenotypic profiles of two pathogenic variants for the hearing loss group. A significant difference in genotype frequencies between cases and controls was observed for the variants c.*84T>C and c.109G>A, while the other seven variants (including c.235delC) were not statistically different between cases and controls. Homozygous mutations in c.109G>A and c.235delC showed different degrees of hearing loss and various shapes of audiogram, mainly sloping and flat. This study was a large-scale population analysis of *GJB2*-related variants (including 4168 participants with normal hearing and 5742 patients with hearing loss), and contained objective pure tone audiometric measurements that could effectively characterize the hearing phenotype.

Consistent with our expectations, we found that the genotype frequencies of the c.109G>A variant in *GJB2* were statistically significantly distributed between cases and controls. The frequency of homozygous genotype in the hearing impaired group was 0.5%, lower than that reported in other Chinese populations in a recent study [17], where c.109G>A homozygous genotypes accounted for 2.6% of patients with hearing loss (there were eleven c.109G>A homozygous genotypes in the eighty-eight patients with mild to moderate hearing loss and fourteen in the eight hundred and fifty-seven patients with severe to profound hearing loss). A possible explanation for this difference could be the different populations included. The age of the subjects in our study was generally much older, which might not exclude the contribution of aging to the hearing loss phenotype. In addition, we observed that the allele frequencies of c.109G>A were approximately equal between the normal and impaired hearing groups, with no significant difference. This might be because most of the alleles were contributed by unaffected heterozygote carriers, as observed in our study where the proportion of c.109G>A heterozygotes approximated between control and case groups (Table 2). Accordingly, it is more appropriate to compare genotype frequencies rather than allele frequencies in case and control analyses to interpret the variants related to autosomal recessive inheritance patterns in the *GJB2* gene [13]. In agreement with several previous studies [10,17,28], we observed that patients with the c.109G>A homozygous genotype displayed different levels of hearing loss. Notably, Chai et al. further investigated potentially pathogenic variants in fourteen patients with severe to profound hearing loss of the c.109G>A homozygous genotype, and they identified another pathogenic variant of *CDH23* coexisting with the c.109G>A homozygous mutation in one patient [17]. Therefore, when c.109G>A homozygous mutations are identified in patients with severe to profound hearing loss, there may be concurrent mutations in other genes or other alternative causes, such as infection or aging [13,17,28]. However, because of the lack of such data, alternative etiologies of c.109G>A homozygous mutations in patients with severe to profound hearing loss were not investigated in our study, and further studies are needed to elucidate them.

The c.235delC variant has been reported to be the most prevalent *GJB2*-associated variant in Chinese patients with hearing loss [4,12,29,30]. In addition, Dai et al. [29] reported that the frequency distributions of 235delC homozygous and heterozygous genotypes in patients with hearing loss varied significantly in different regions of China, ranging from 0% to 14.7% for the homozygous genotype and 1.7% to 16.1% for the heterozygous genotype. In general, the 235delC variant is more commonly found in northern China [12]. In our study, we observed two hearing-impaired patients with c.235delC homozygous genotypes, who showed moderate and severe to profound hearing loss, respectively. Participants in the DFTJ cohort were retired employees of Dongfeng Motor Corporation who had normal hearing before beginning work. However, it should be noted that individuals with severe hearing loss would not be hired. This restriction on hearing performance might explain the inconsistent frequency of c.235delC homozygous mutations observed in our study compared to other studies. The 235delC variant in the *GJB2* gene causes a frameshift mutation that prematurely terminates translation and produces a non-functional gap junction protein [31,32]. Patients with the c.235delC homozygous genotype usually present with severe to profound prelingual hearing loss [20,33,34]. However, other studies have shown that patients with the c.235delC homozygous mutation also show mild to moderate hearing phenotypes [30,34]. Consistent with these findings, one of the patients in our study with the c.235delC homozygous genotype had moderate hearing loss, suggesting that the hearing phenotype of patients with the c.235delC homozygous mutation was diverse. 

We found that the genotype and allele frequencies of c.*84T>C were statistically different between the normal and impaired hearing groups. The c.*84T>C is located in the 3′UTR region of the *GJB2* gene and has been identified as a benign variant. Technological advances in sequencing and many other assays have facilitated the elucidation of the non-coding genome [35]. Many noncoding elements have been found to be linked to hearing loss, and the identification of non-coding variants in hearing loss is expected to improve diagnostic rates [35]. The genotype and allele frequencies of the c.*84T>C variant reported in our study may provide a molecular epidemiologic basis for future studies on non-coding region variants. In addition, two previous studies reported a significant association of the c.*84T>C with noise-induced hearing loss [36]. We observed that the allele frequencies of four benign variants (c.79G>A, c.341A>G, c.368C>A, and c.608T>C) were similar between cases and controls, whereas another study reported that their allele frequencies were greater in controls than in patients [5].

In agreement with a previous study [10], the audiogram shapes of the c.109G>A and c.235delC variants presented various audiometric configurations in this study, with a dominance of sloping and flat types. Similarly, Liu et al. found that audiogram shapes in most patients with *GJB2*-related hearing loss were generally residual, sloping, or flat, but similar to those observed in hearing impaired patients without the *GJB2* mutation, suggesting a possible poor correlation between the type of *GJB2* mutation and audiogram shape [20].

The limitations of this study should also be considered. First, the participants were from a single cohort and might not be representative of the general Chinese population, which limited the generalizability of the results. Second, other pathogenic variants in GJB2 were not included in the study. Finally, this study also did not consider the case of compound heterozygotes in the GJB2 gene or in combination with other genes, which requires further research.

## 5. Conclusions

To sum up, we observed a statistically significant distribution of genotype frequencies of the *GJB2* variants c.*84T>C and c.109G>A between cases and controls in a Chinese population in the large-scale DFTJ cohort, but not for the c.235delC variant. Patients with c.109G>A and c.235delC homozygous mutations exhibited diverse hearing loss severities and audiogram shapes. These findings could provide evidence for genetic diagnosis and counseling for hearing loss.

## Figures and Tables

**Figure 1 genes-14-02007-f001:**
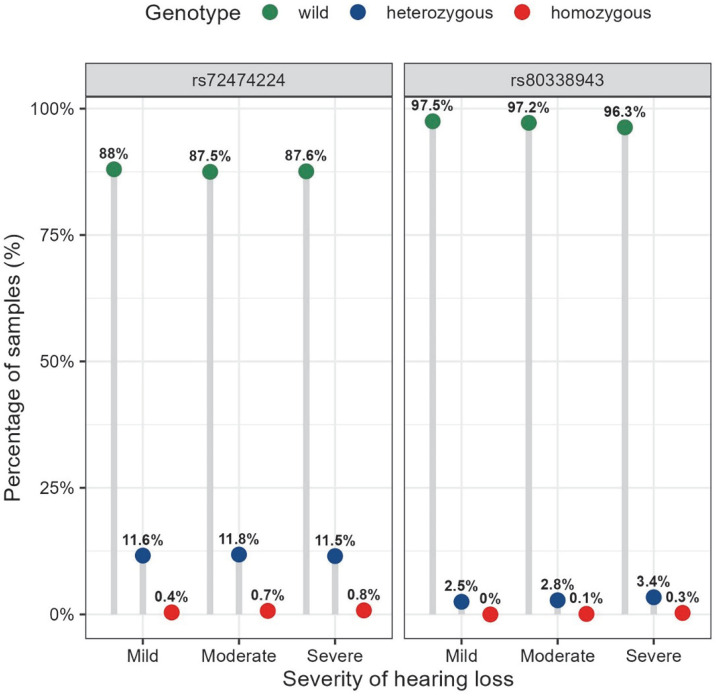
Frequency distribution of the genotypes for rs72474224 (c.109G>A) and rs80338943 (c.235delC) in the GJB2 gene across hearing loss severity groups. Note: The genotypes of wild, heterozygous, and homozygous for rs72474224 and rs80338943 are: C/C, T/C, and T/T; and AG/AG, A/AG, and A/A, respectively. The distribution of the genotypes of rs72474224 and rs80338943 among different hearing loss severity groups was tested using Fisher’s exact test, with *p* = 0.369 and *p* = 0.091, respectively.

**Figure 2 genes-14-02007-f002:**
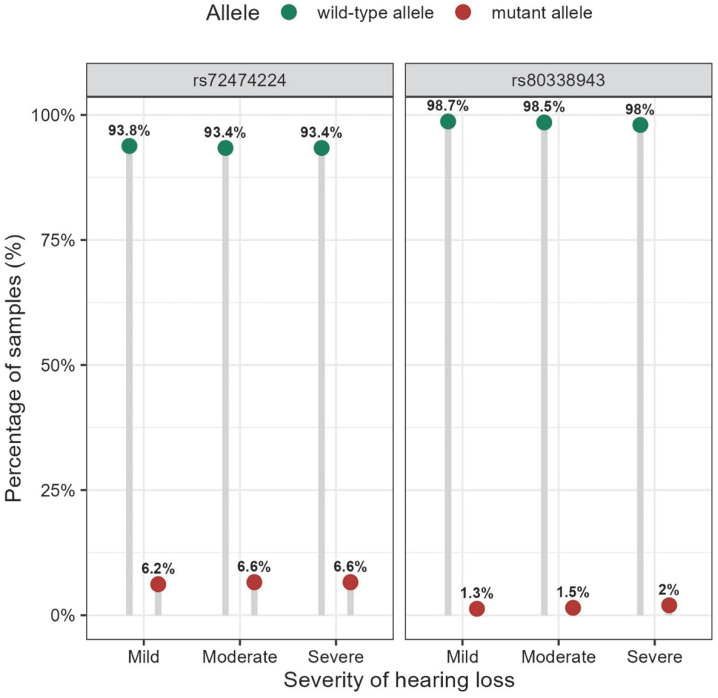
Frequency distribution of the alleles for rs72474224 (c.109G>A) and rs80338943 (c.235delC) in the GJB2 gene across hearing loss severity groups. Note: The wild-type and mutant alleles for rs72474224 and rs80338943 are as follows: C and T; and AG as well as A, respectively. The distribution of the alleles of rs72474224 and rs80338943 in different hearing loss severity groups was tested using the chi-squared test, and *p* values were: *p* = 0.274; and *p* = 0.661, respectively.

**Figure 3 genes-14-02007-f003:**
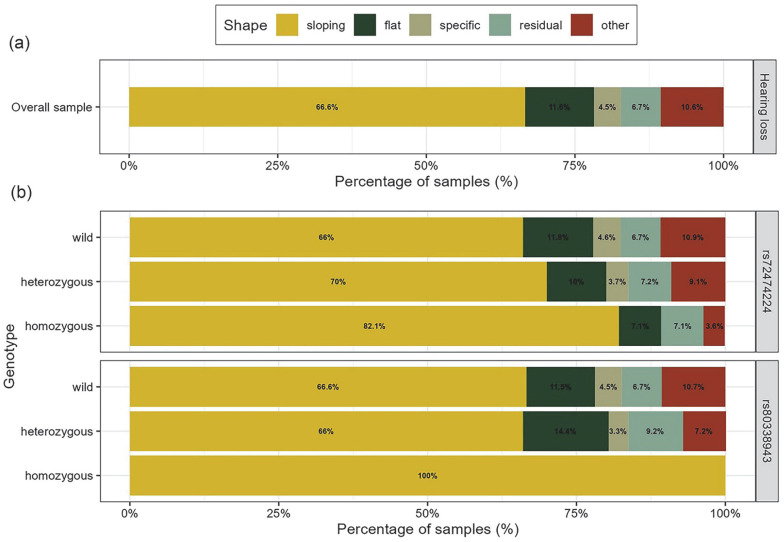
The audiogram shapes (**a**) in 5742 participants with hearing loss and (**b**) in the different genotypes of rs72474224 (c.109G>A) and rs80338943 (c.235delC). Note: The genotypes of wild, heterozygous, and homozygous for rs72474224 and rs80338943 are: C/C, T/C and T/T; and AG/AG, A/AG, and A/A, respectively. The distribution of the types of audiogram in different genotypes of rs72474224 and rs80338943 was tested using Fisher’s exact test, and *p* values were: *p* = 0.324, and *p* = 0.576, respectively.

**Table 1 genes-14-02007-t001:** Characteristics of overall participants.

Characteristic	Total (n = 9910)
Age, mean (SD), years	67.2 (7.6)
Age group (%)	
<60	1548 (15.6)
60–69	4765 (48.1)
70–79	3054 (30.8)
≥80	543 (5.5)
Gender (%)	
Female	5197 (52.4)
Male	4713 (47.6)
Severity of hearing loss (%)	
Normal	4168 (42.1)
Mild	3806 (38.4)
Moderate	1581 (16.0)
Severe-profound	355 (3.6)

Abbreviations: SD, standard deviation. Data were expressed as mean (SD) for continuous variables and number (percentage) for categorical variables.

**Table 2 genes-14-02007-t002:** Frequency distribution of genotypes and alleles of single nucleotide polymorphisms in the *GJB2* gene.

SNPs	Nucleotide Change	Protein Change	Clinical Significance	Overall (n = 9910)	Hearing Loss		*p* Value
					Control Group (n = 4168)	Patients Group (n = 5742)	
rs111033222	c.11G>A	p.G4D	Conflicting				
Genotype	C/C			9854 (99.4)	4143 (99.4)	5711 (99.5)	0.797 ^a^
	T/C			56 (0.6)	25 (0.6)	31 (0.5)	
Allele	C			19,764 (99.7)	8311 (99.7)	11,453 (99.7)	0.797 ^a^
	T			56 (0.3)	25 (0.3)	31 (0.3)	
rs2274084	c.79G>A	p.V27I	Benign				
Genotype	C/C			4391 (44.3)	1854 (44.5)	2537 (44.2)	0.680 ^a^
	T/C			4518 (45.6)	1906 (45.7)	2612 (45.5)	
	T/T			1001 (10.1)	408 (9.8)	593 (10.3)	
Allele	C			13,300 (67.1)	5614 (67.3)	7686 (66.9)	0.546 ^a^
	T			6520 (32.9)	2723 (32.7)	3798 (33.1)	
rs3751385	c.*84T>C	-	Benign				
Genotype	A/A			2157 (21.8)	850 (20.4)	1307 (22.8)	**0.017** ^a^
	A/G			4985 (50.3)	2142 (51.4)	2843 (49.5)	
	G/G			2768 (27.9)	1176 (28.2)	1592 (27.7)	
Allele	A			9299 (46.9)	3842 (46.1)	5457 (47.5)	**0.048** ^a^
	G			10,521 (53.1)	4494 (53.9)	6027 (52.5)	
rs72474224	c.109G>A	p.V37I	Pathogenic				
Genotype	C/C			8718 (88.0)	3674 (88.1)	5044 (87.8)	**<0.001** ^b^
	T/C			1161 (11.7)	491 (11.8)	670 (11.7)	
	T/T			31 (0.3)	3 (0.1)	28 (0.5)	
Allele	C			18,597 (93.8)	7839 (94.0)	10,758 (93.7)	0.313 ^a^
	T			1223 (6.2)	497 (6.0)	726 (6.3)	
rs80338943	c.235delC	p.Leu79fs	Pathogenic				
Genotype	AG/AG			9665 (97.5)	4078 (97.8)	5587 (97.3)	0.118 ^b^
	A/AG			243 (2.5)	90 (2.2)	153 (2.7)	
	A/A			2 (0.02)	0 (0.0)	2 (0.03)	
Allele	AG			19,573 (98.8)	8246 (98.9)	11,327 (98.6)	0.083 ^a^
	A			247 (1.2)	90 (1.1)	157 (1.4)	
rs2274083	c.341A>G	p.E114G	Benign/Likely benign				
Genotype	T/T			5284 (53.3)	2249 (54.0)	3035 (52.9)	0.423 ^a^
	C/T			4025 (40.6)	1678 (40.3)	2347 (40.9)	
	C/C			601 (6.1)	241 (5.8)	360 (6.3)	
Allele	T			14,593 (73.6)	6176 (74.1)	8417 (73.3)	0.216 ^a^
	C			5227 (26.3)	2160 (25.9)	3067 (26.7)	
rs111033188	c.368C>A	p.T123N	Benign/Likely benign				
Genotype	G/G			9828 (99.2)	4134 (99.2)	5694 (99.2)	0.965 ^b^
	T/G			80 (0.8)	33 (0.8)	47 (0.8)	
	T/T			2 (0.02)	1 (0.02)	1 (0.02)	
Allele	G			19,736 (99.6)	8301 (99.6)	11,435 (9.6)	0.942 ^a^
	T			84 (0.4)	35 (0.4)	49 (0.4)	
rs397516878	c.571T>C	p.F191L	Uncertain				
Genotype	A/A			9899 (99.9)	4163 (99.9)	5736 (99.9)	0.871 ^b^
	A/G			11 (0.1)	5 (0.1)	6 (0.1)	
Allele	A			19,809 (99.9)	8331 (99.9)	11,478 (99.95)	0.871 ^b^
	G			11 (0.1)	5 (0.1)	6 (0.05)	
rs76838169	c.608T>C	p.I203T	Benign				
Genotype	A/A			9298 (93.8)	3929 (94.3)	5369 (93.5)	0.289 ^a^
	A/G			598 (6.0)	234 (5.6)	364 (6.3)	
	G/G			14 (0.1)	5 (0.1)	9 (0.2)	
Allele	A			19,194 (96.8)	8092 (97.1)	11,102 (96.7)	0.122 ^a^
	G			626 (3.2)	244 (2.9)	382 (3.3)	

Abbreviations: SNPs, single nucleotide polymorphisms. ^a^ *p* values were estimated via chi-square test; ^b^ *p* values were estimated via Fisher’s exact test. *p* values in bold indicated statistical significance.

**Table 3 genes-14-02007-t003:** Characteristics and pure tone threshold of participants with the homozygous genotype of rs72474224 (c.109G>A) (n = 31).

Hearing Loss	Medical ID	Age (Years)	Gender	Better Ear (dB)					
				0.5 kHz	1 kHz	2 kHz	4 kHz	8 kHz	PTA 0.5–4 kHz
Normal	106846	58	Female	15	15	5	10	10	11.25
Normal	105802	59	Female	15	5	25	40	60	21.25
Normal	90678	68	Male	25	15	20	25	30	21.25
Mild	78353	60	Male	20	25	25	40	65	27.5
Mild	70091	61	Female	25	25	20	50	45	30
Mild	70740	61	Female	30	30	30	40	50	32.5
Mild	11011	63	Female	15	15	50	50	70	32.5
Mild	4888	64	Female	25	30	40	45	75	35
Mild	60195	64	Female	25	20	30	40	70	28.75
Mild	107326	64	Female	25	30	35	55	65	36.25
Mild	54382	66	Male	20	30	40	45	95	33.75
Mild	101181	66	Male	30	35	35	40	65	35
Mild	11321	67	Male	20	30	35	60	85	36.25
Mild	67746	69	Female	25	25	25	45	90	30
Mild	39311	71	Male	15	15	30	60	25	30
Mild	59571	74	Female	15	15	40	60	70	32.5
Mild	53680	76	Male	20	20	30	35	90	26.25
Moderate	54632	56	Female	35	35	35	60	90	41.25
Moderate	79049	61	Female	35	40	50	45	55	42.5
Moderate	16099	67	Male	20	30	60	70	85	45
Moderate	28532	67	Male	35	35	60	70	80	50
Moderate	90158	68	Male	40	50	50	65	55	51.25
Moderate	94371	68	Male	30	45	65	75	100	53.75
Moderate	42832	77	Male	15	30	60	70	90	43.75
Moderate	100741	77	Female	20	40	55	50	80	41.25
Moderate	116341	78	Male	40	40	50	60	85	47.5
Moderate	107144	79	Male	20	25	55	70	90	42.5
Moderate	5826	80	Male	40	50	65	85	90	60
Severe	56456	64	Male	50	55	65	90	110	65
Severe	115724	73	Male	60	70	70	75	110	68.75
Severe	93245	74	Male	60	60	70	70	100	65

Abbreviations: PTA, pure tone audiometry. Note: The homozygous genotype of rs72474224 (c.109G>A) is T/T.

## Data Availability

Requests may be addressed to the corresponding authors.

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
