# Peer review of "Genotypic and Allelic Frequencies of GJB2 Variants and Features of Hearing Phenotypes in the Chinese Population of the Dongfeng-Tongji Cohort"

_genes, 2023, doi:10.3390/genes14112007_

Round 1
Reviewer 1 Report
Comments and Suggestions for Authors
In this study, the authors sought to determine the allele frequencies and distribution of 9 GJB2 variants from the Dongfeng-Tongji cohort. Two of the nine variants, p.V37I and c.*84T>C showed statistically significant difference between the controls and case. The pathogenic variant, p.V37I group exhibited varying degrees of hearing loss and audiogram shapes. Overall, this is a nicely presented study. The introduction is succinct, methodology is detailed enough. The presentation of the results is apt. Conclusions are supported by the results presented. I have some minor comments.
1. The sentence (lines 40-43) doesn't seem to fit in the introduction. It would be better to remove it.
2. Section 3.4 doesn't add any additional information to the manuscript. No new variants are reported in this study.
Author Response
In this study, the authors sought to determine the allele frequencies and distribution of 9 GJB2 variants from the Dongfeng-Tongji cohort. Two of the nine variants, p.V37I and c.*84T>C showed statistically significant difference between the controls and case. The pathogenic variant, p.V37I group exhibited varying degrees of hearing loss and audiogram shapes. Overall, this is a nicely presented study. The introduction is succinct, methodology is detailed enough. The presentation of the results is apt. Conclusions are supported by the results presented. I have some minor comments.
1. The sentence (lines 40-43) doesn't seem to fit in the introduction. It would be better to remove it.
Response: We thank the reviewer for the suggestion to improve the paper. We have removed this sentence from the Introduction in the revised manuscript.
2. Section 3.4 doesn't add any additional information to the manuscript. No new variants are reported in this study.
Response: Thank you for your kind comment. Section 3.4 has been removed from the revised manuscript.

Reviewer 2 Report
Comments and Suggestions for Authors
In the manuscript” Genotypic and allelic frequencies of GJB2 variants and features hearing loss phenotypes in the Chinese population of Dongfeng-Tongji cohort” the authors have performed a detailed statistical analysis of hearing phenotypes and nine variants of GJB2 gene which are present most commonly in their population. The paper can significantly contribute to the field of study, it is well structured and the language used in the manuscript is clear and concise. The reviewer doesn’t have any major concerns, hence the reviewer recommend the manuscript for publication after following minor corrections:
· The reviewer suggests authors to provide more details about the genome wide association study performed, in the materials & methods section as it is not clear whether it was performed exclusively for the current study or it is part of some previous work/publication?
· The finding of Figures 1 & 2 are not explained well in the manuscript, please add some more details about the findings and the significance of the both analyses.
· In the table # 1, it’s not clear what “y” is representing? Please write the full form and also describe that the values in brackets are showing the “percentage”.
· In the lines 133-134, please describe the mean age of controls and patients of the study separately for the better comparison of their disease symptoms & age of the both groups.
· In Table # 3 legends, please write down the respective genotypes.
· In the Table # 2, please highlight or bold the P-values which were found significant.
· In the line 131, please change “sample” into “samples”.
· In the line 217, please change “the grey-block color represent uncertainty” into “the grey-block color represents variant of uncertain significance”.
Author Response
In the manuscript “Genotypic and allelic frequencies of GJB2 variants and features hearing loss phenotypes in the Chinese population of Dongfeng-Tongji cohort” the authors have performed a detailed statistical analysis of hearing phenotypes and nine variants of GJB2 gene which are present most commonly in their population. The paper can significantly contribute to the field of study, it is well structured and the language used in the manuscript is clear and concise. The reviewer doesn’t have any major concerns, hence the reviewer recommend the manuscript for publication after following minor corrections:
1. The reviewer suggests authors to provide more details about the genome wide association study performed, in the materials & methods section as it is not clear whether it was performed exclusively for the current study or it is part of some previous work/publication?
Response: We thank the reviewer for this very relevant comment. More detailed information on the genome wide association study has been included in section 2.3 of the Materials and Methods. The text now reads (please refer to lines 123-128 in red in the revised manuscript):
“In brief, these SNPs were successfully genotyped after quality control, excluding any SNPs with missing call rate >5%, minor allele frequency <1% and P value of Hardy-Weinberg equilibrium <10-5. There were 703302 SNPs in the Illumina dataset and 549196 in the Affymetrix dataset. Genotypes of markers in the Illumina and Affymetrix datasets were imputed separately using the 1000 Genomes Project ALL Phase 3 Integrated Release Version 5 haplotypes as a reference panel.”
2. The finding of Figures 1 & 2 are not explained well in the manuscript, please add some more details about the findings and the significance of the both analyses.
Response: We thank the reviewer for the suggestion. As shown in Figures 1 and 2, the x-axis represents the severity of hearing loss. Accordingly, Figures 1 and 2 show the correlation of different genotypes (genotype-phenotype correlation) and alleles (allele-phenotype correlation) with the severity of hearing loss for c.109G>A (rs72474224) and c.235delC (rs80338943) variants, respectively. Figure 1 shows that the frequencies of homozygous genotypes increased with hearing threshold for the two variants (described in lines 163-165 and 193-194, respectively), but no significant differences were observed between groups, suggesting that the c.109G>A (described in lines 251-252 in the Discussion section) and the c.235delC (described in lines 281-282 in the Discussion section) variants exhibited different degrees of hearing loss.
Regarding Figure 2, although the c.235delC variant showed an increasing trend in allele frequency, the c.109G>A variant did not, and no statistical difference in the distribution of allele frequencies was found (described in lines 166-168 and 194-196, respectively).
The results in Figures 1 and 2 further suggest that it is more appropriate to compare genotype frequencies rather than allele frequencies to interpret variants in the GJB2 gene associated with an autosomal recessive inheritance pattern (described in lines 248-250 in the Discussion section), namely that the hearing phenotype of the GJB2 variants is highly dependent on genotype (described in lines 55-56 in the Introduction section).
3. In the table # 1, it’s not clear what “y” is representing? Please write the full form and also describe that the values in brackets are showing the “percentage”.
Response: We thank the reviewer for pointing this out. In Table 1, “y” represents the unit of age, i.e., years. We have corrected it. Below the table, we have provided the description of the data in the table as follows (please refer to lines 146-147 in red in the revised manuscript):
“Data were expressed as mean (SD) for continuous variables and number (percentage) for categorical variables.”
4. In the lines 133-134, please describe the mean age of controls and patients of the study separately for the better comparison of their disease symptoms & age of the both groups.
Response: We thank the reviewer for the suggestion. We have added a brief explanation, as suggested. The text now reads (please refer to lines 143-144 in red in the revised manuscript):
“In addition, the mean ages of the controls and patients were 63.9 (7.0) and 69.7 (7.1) years, respectively.”
5. In Table # 3 legends, please write down the respective genotypes.
Response: We thank the reviewer for the suggestion. We have added a short legend to Table 3, text as follows (please refer to lines 175-176 in red in the revised manuscript):
“Note: The homozygous genotype of rs72474224 (c.109G>A) is C/C.”
6. In the Table # 2, please highlight or bold the P-values which were found significant.
Response: We thank the reviewer for the suggestion. We have bolded the significant P values in Table 2 and added the corresponding legend with the text as follows (please refer to line 172 in red in the revised manuscript):
“P values in bold indicated statistical significance.”
7. In the line 131, please change “sample” into “samples”.
Response: We thank the reviewer for the suggestion. We have corrected it (please refer to line 139 in red in the revised manuscript).
8. In the line 217, please change “the grey-block color represent uncertainty” into “the grey-block color represents variant of uncertain significance”.
Response: We thank the reviewer for the suggestion. However, following the suggestion of reviewer 1, we have removed Section 3.4 from the revised manuscript. Please also see our response to comment 2 made by reviewer 1.

Reviewer 3 Report
Comments and Suggestions for Authors
Describe the molecular basis according to gene reviews (https://www.ncbi.nlm.nih.gov /books/NBK1272/).Large deletions involving GJB2 are described, they are tested? Also, c.235delC is described as a mosaic and maternal UPD
Describe the variants in the same way genomic vs protein c.235delC and c.109G>A (not p.V37I.
In methods: the sentence For the purpose of genetic analysis related to hearing loss, nine common GJB287 mutations in the Chinese population were selected move to 2.3 Genotyping
The 4168 people that did not present hearing loss were used as a control group?
GJB2-related autosomal recessive nonsyndromic hearing loss (GJB2-AR NSHL) is the most common genetic cause of congenital (present at birth) and the cohort is retired people? Authors in the discussion described that people severe hearing loss not be hired, infantile hearing loss in the cohort is excluded? The age of individuals contributes in the hearing loss
Why authors tested Benign and likely benign variants, are they suggesting that carrier of a variant in GBJ2 could be responsible for a dominant inheritance? The variants were tested and only two are pathogenic. Premature termination codons (PTC), such as c.35delG, c.167delT,
Authors are suggesting that variant c.*84T>C considered benign are responsible of hearing loss in carrier status? Other genetic should be excluded (test other genes) using exome or customized gene panel no an array of SNPs
Inheritance of GBJ2 is described as a digenic variant with other genes not tested in this study
Compound heterozygotes of other pathogenic variants in GJB2 are not considered because only two pathogenic variants are tested. A higher penetrance for compound heterozygous genotypes than p.Met34Thr or p.Val37Ile homozygosity is suggested by Shen et al [18] so the authors in line 54 described: Currently, p.V37I is 54 considered a pathogenic mutation with incomplete penetrance [18].In the other hand in this study there are 31 people with homozygous variant in V37I
One other objectives of this paper is: to investigate the audiogram shapes in patients with hearing loss under different genotypes of p.V37I and c.235delC mutations in the GJB2 gene. May be the results should be focused on the 31 homozygotes considered as a pathogenic?
Author Response
1. Describe the molecular basis according to gene reviews (https://www.ncbi.nlm.nih.gov /books/NBK1272/). Large deletions involving GJB2 are described, they are tested? Also, c.235delC is described as a mosaic and maternal UPD
Response: We thank the reviewer for the very insightful comment. Regarding large deletions involving GJB2, this is indeed a very relevant point to be considered. Unfortunately, we did not have access to such additional data in this study. However, we acknowledged that this was a limitation of our study. To make it clearer, we have included the following sentence in the limitation section (please refer to lines 305-306 in red in the revised manuscript):
“Second, other pathogenic variants in GJB2 were not included in the study.”
2. Describe the variants in the same way genomic vs protein c.235delC and c.109G>A (not p.V37I.
Response: We thank the reviewer for the suggestion. We have corrected all texts in the revised manuscript to describe the variants in the same way (i.e., the genomic format).
3. In methods: the sentence For the purpose of genetic analysis related to hearing loss, nine common GJB287 mutations in the Chinese population were selected move to 2.3 Genotyping
Response: We thank the reviewer for the suggestion. We have moved this sentence “For the purpose of genetic analysis related to hearing loss, nine common GJB2 mutations in the Chinese population were selected” to section 2.3 Genotyping (please refer to lines 116-117 in red in the revised manuscript).
4. The 4168 people that did not present hearing loss were used as a control group?
Response: We thank the reviewer for the suggestion. As presented in Table 2, the 4168 people without hearing loss were used as a control group.
5. GJB2-related autosomal recessive nonsyndromic hearing loss (GJB2-AR NSHL) is the most common genetic cause of congenital (present at birth) and the cohort is retired people? Authors in the discussion described that people severe hearing loss not be hired, infantile hearing loss in the cohort is excluded? The age of individuals contributes in the hearing loss
Response: We thank the reviewer for the suggestion. As described in Section 2.1 of this study, the DFTJ cohort is a prospective study based on retired employees of Dongfeng Motor Corporation. Thus, infants with hearing loss were not included in the DFTJ cohort. We completely agree with the reviewer that the age of individuals may contribute to hearing loss, which we have discussed in the manuscript (please refer to lines 242-243 in red in the Discussion section) as follows:
“The age of the subjects in our study was generally much older, which might not exclude the contribution of aging to the hearing loss phenotype.”
6. Why authors tested Benign and likely benign variants, are they suggesting that carrier of a variant in GBJ2 could be responsible for a dominant inheritance? The variants were tested and only two are pathogenic. Premature termination codons (PTC), such as c.35delG, c.167delT,
Response: We thank the reviewer for the suggestion.
Although only nine variants (including the benign and likely benign variants) were included in this study, our major objective was to evaluate the spectrum and frequency of GJB2 variants in the Chinese population as comprehensively as possible. We acknowledge that pathogenic variants are more worthy of attention. However, this study was a large-scale population-based analysis of GJB2-associated variants, and the identified spectrum and frequency of GJB2 variants (including benign and likely benign variants) could provide data support for future research and clinical genetic counseling. However, we acknowledged that the lack of inclusion of other variants in GJB2 was a limitation of our study. To make this clearer, we have added the following sentence to the limitation section (please refer to lines 305-306 in red in the revised manuscript):
“Second, other pathogenic variants in GJB2 were not included in the study.”
7. Authors are suggesting that variant c.*84T>C considered benign are responsible of hearing loss in carrier status? Other genetic should be excluded (test other genes) using exome or customized gene panel no an array of SNPs
Response: We thank the reviewers for this very relevant comment.
As described in the response to the prior comment, this study aimed to evaluate the spectrum and frequency of GJB2 variants in the Chinese population as comprehensively as possible, including the c.*84T>C variant considered benign. Regarding the c.*84T>C variant, only two previous studies reported the association of c.*84T>C with noise-induced hearing loss (described in lines 291-292 in the Discussion section). It is yet unclear whether the c.*84T>C variation is responsible for the hearing loss, and more research may be required. The genotype and allele frequencies of the c.*84T>C variant found in our study may provide epidemiological data support for future studies.
Regarding the reviewer's suggestion that “Other genetic should be excluded (test other genes) using exome or customized gene panel no an array of SNPs”, we completely agree with the importance of this type of study. However, we currently have no available data on this, and it will be the focus of our future studies.
8. Inheritance of GBJ2 is described as a digenic variant with other genes not tested in this study
Response: We thank the reviewers for this very relevant comment. We agree that the inheritance of GBJ2 is described as a digenic variant with other genes. For example, as we described in the Discussion section (please refer to lines 252-256 in red in the revised manuscript), Chai et al. identified another pathogenic variant of CDH23 coexisting with the c.109G>A homozygous mutation in one of fourteen patients with severe to profound hearing loss of the c.109G>A homozygous genotype. In this study, the patients with hearing loss found to have homozygous mutations in the GBJ2 gene might also be involved in mutations in other genes, such as compound heterozygous mutations in combination with other genes. However, in this study, our focus was on the variants in the GJB2 gene, and other genes were not tested. On the other hand, as described in our study (please refer to lines 259-261 in red in the Discussion section), further studies are needed to test other related genes for a more comprehensive understanding of the etiologies of hearing loss. However, we acknowledged that this was a limitation of our study. To make this clearer, we have added the following sentence to the limitation section (please refer to lines 306-308 in red in the revised manuscript):
“Finally, this study also did not consider the case of compound heterozygotes in the GJB2 gene or in combination with other genes, which requires further research.”
9. Compound heterozygotes of other pathogenic variants in GJB2 are not considered because only two pathogenic variants are tested. A higher penetrance for compound heterozygous genotypes than p.Met34Thr or p.Val37Ile homozygosity is suggested by Shen et al [18] so the authors in line 54 described: Currently, p.V37I is 54 considered a pathogenic mutation with incomplete penetrance [18].In the other hand in this study there are 31 people with homozygous variant in V37I
Response: We thank the reviewers for this very relevant comment. As mentioned in the previous comment, this study did not consider the case of compound heterozygotes, either in combination with other pathogenic variants of the GJB2 gene or other genes, which requires further research.
10. One other objectives of this paper is: to investigate the audiogram shapes in patients with hearing loss under different genotypes of p.V37I and c.235delC mutations in the GJB2 gene. May be the results should be focused on the 31 homozygotes considered as a pathogenic?
Response: We thank the reviewers for this very relevant comment. Yes, the correlation of the homozygous genotype of the p.V37I variant with the hearing phenotype was of interest to us. Therefore, we have not only depicted the audiogram shapes in patients with hearing loss under different genotypes of the p.V37I variant (Figure 3 a), but also presented the detailed pure tone thresholds of the 31 patients with the homozygous genotype in Table 3.

Reviewer 4 Report
Comments and Suggestions for Authors
Dear authors,
thanks for you manuscript describing the genotypev- audiological phenotype correlations in a large population in China. Your article is well written, the research atim is clearly formulated, the methods are well described and the results are well presented.
There are only a few modifications I would like to ask you for:
1. As mention by yourselves in the discussion the audiological characteristics of hearing loss are most likely affected by age. And as a matter of fact, your cohort is an elderly population (sixty-plus). Therefore, please add this information "elderly or sixty-plus" to your title.
2. You should also add it to your keywords, eg. elderly
3. lines 75 ff: Please give more information about the Dongfeng-Tongju cohort. Is the whole cohort representative for elderly adults in China? Are people with congenital hearing loss excluded? Is there any reason to believe that due to the kind of work in the motor corporation there is nore noice induced hearing loss as compared to the total Chinese population?
4. Please give reasons for the reduction of 11.513 to your (almost) final sample of 9912.
5. Any reasons for the PTA of 60 (rather than 70) for severe-profound hearing loss?
6. line 194: I assume that "sloping" should be prefered over "sloped"
7. Discussion: line 219: please add that your population was sixty-plus
8. Please add a limitations section including the restriction to the group of the elderly and a possible selection bias of your cohort that might limit generalization of your findings for the total population.
Author Response
Dear authors,
thanks for you manuscript describing the genotype-audiological phenotype correlations in a large population in China. Your article is well written, the research atim is clearly formulated, the methods are well described and the results are well presented.
There are only a few modifications I would like to ask you for:
1. As mention by yourselves in the discussion the audiological characteristics of hearing loss are most likely affected by age. And as a matter of fact, your cohort is an elderly population (sixty-plus). Therefore, please add this information "elderly or sixty-plus" to your title.
Response: We thank the reviewer for the suggestion to improve the paper, but we argue that there were 15.6% of people aged under 60 years in the study (as shown in Table 1). Based on the fact that this study was not restricted to the elderly population, we have decided not to change the title.
2. You should also add it to your keywords, eg. elderly
Response: We thank the reviewer for this very relevant comment. As explained in response to the first comment, we have decided not to add it to the keywords.
3. lines 75 ff: Please give more information about the Dongfeng-Tongju cohort. Is the whole cohort representative for elderly adults in China? Are people with congenital hearing loss excluded? Is there any reason to believe that due to the kind of work in the motor corporation there is more noise induced hearing loss as compared to the total Chinese population?
Response: We thank the reviewer for the suggestion. We agree that information about the Dongfeng Tongji (DFTJ) cohort has not been sufficiently described in the Materials and Methods section. To make this clearer, we have included the following sentence in the Materials and Methods section (please refer to lines 76-79 in red in the revised manuscript):
“Information on demographic characteristics, lifestyle, health status, and medical history was collected by trained interviewers using a standardized questionnaire. In addition, participants performed physical examinations and provided samples of peripheral venous blood after an overnight fast.”
As explained in response to the first comment, the cohort was not restricted to the elderly population, and is therefore not representative of older people in China.
Participants in the DFTJ cohort were recruited from Dongfeng Motor Corporation. It should be noted that individuals with severe hearing loss would not be recruited. Therefore, patients with congenital hearing loss were not included.
In a 2017 study based on the DFTJ cohort [1], Wang et al. found that participants with a duration of noise exposure ≥20 years were associated with a higher risk of hearing loss (OR = 1.45, 95% CI = 1.28–1.65) compared to participants without occupational noise exposure, which was not found in participants with noise exposure durations of 1–9 years and 10–19 years. Furthermore, among the 11,196 participants included in the analysis, only 16.0% had a noise exposure duration of ≥20 years. According to the findings of Wang et al., a small proportion of the DFTJ cohort may be more likely to have noise-induced hearing loss.
References
- Wang, D.; Wang, Z.; Zhou, M.; et al. The combined effect of cigarette smoking and occupational noise exposure on hearing loss: evidence from the Dongfeng-Tongji Cohort Study. Sci Rep 2017,7(1), 11142.
4. Please give reasons for the reduction of 11.513 to your (almost) final sample of 9912.
Response: We thank the reviewer for the suggestion. We agree that this has not been sufficiently described in the Materials and Methods section, and we have therefore included the following in the text (please refer to lines 88-90 in red in the revised manuscript):
“With the exclusion of 57 participants with invalid audiometric data and 1544 participants who lacked genotype and failed quality control, this study included 9912 participants with complete data on nine GJB2 mutations and audiometry.”
5. Any reasons for the PTA of 60 (rather than 70) for severe-profound hearing loss?
Response: We thank the reviewer for the suggestion. Following the 1991 WHO criteria for grades of hearing loss (Ref. 29 of the manuscript), we defined severe to profound hearing loss as the PTA 0.5–4 kHz greater than 60 dB HL.
References
- Report of the Informal Working Group on Prevention of Deafness and Hearing Impairment Programme Planning, Geneva, 18-21 June 1991. Available online: https://apps.who.int/iris/handle/10665/58839
6. line 194: I assume that "sloping" should be prefered over "sloped"
Response: We thank the reviewer for pointing this out. We have corrected the word “sloped” to “sloping” in line 207 of the revised manuscript.
7. Discussion: line 219: please add that your population was sixty-plus
Response: We thank the reviewer for the suggestion. As explained in response to the first comment, we have decided not to add it to line 223 of the revised manuscript.
8. Please add a limitations section including the restriction to the group of the elderly and a possible selection bias of your cohort that might limit generalization of your findings for the total population.
Response: We thank the reviewer for this very relevant comment. We have added a limitation section in the revised manuscript. The text now reads (please refer to lines 303-308 in red in the Discussion section):
“The limitations of this study should also be considered. First, the participants were from a single cohort and might not be representative of the general Chinese population, which limited the generalizability of the results. Second, other pathogenic variants in GJB2 were not included in the study. Finally, this study also did not consider the case of compound heterozygotes in the GJB2 gene or in combination with other genes, which requires further research.”

Round 2
Reviewer 3 Report
Comments and Suggestions for Authors
no comments